# Microvesicles in Cancer: Small Size, Large Potential

**DOI:** 10.3390/ijms21155373

**Published:** 2020-07-28

**Authors:** Kerstin Menck, Suganja Sivaloganathan, Annalen Bleckmann, Claudia Binder

**Affiliations:** 1Department of Medicine A, Hematology, Oncology, and Pneumology, University Hospital Münster, 48149 Münster, Germany; Kerstin.Menck@ukmuenster.de (K.M.); Suganja.Sivaloganathan@ukmuenster.de (S.S.); annalen.bleckmann@ukmuenster.de (A.B.); 2Department of Hematology/Medical Oncology, University Medical Center Göttingen, 37075 Göttingen, Germany

**Keywords:** microvesicles, biomarker, cancer, tumor microenvironment, therapy

## Abstract

Extracellular vesicles (EV) are secreted by all cell types in a tumor and its microenvironment (TME), playing an essential role in intercellular communication and the establishment of a TME favorable for tumor invasion and metastasis. They encompass a variety of vesicle populations, among them the well-known endosomal-derived small exosomes (Exo), but also larger vesicles (diameter > 100 nm) that are shed directly from the plasma membrane, the so-called microvesicles (MV). Increasing evidence suggests that MV, although biologically different, share the tumor-promoting features of Exo in the TME. Due to their larger size, they can be readily harvested from patients’ blood and characterized by routine methods such as conventional flow cytometry, exploiting the plethora of molecules expressed on their surface. In this review, we summarize the current knowledge about the biology and the composition of MV, as well as their role within the TME. We highlight not only the challenges and potential of MV as novel biomarkers for cancer, but also discuss their possible use for therapeutic intervention.

## 1. Introduction

Tumor development is a multistep process that is accompanied by various cellular reprogramming events in which cells acquire the hallmarks of cancer cells to gain and sustain abnormal growth and invasive capacity [1]. The complex process of tumor formation and spreading not only depends on the tumor cells themselves but also requires rewiring of the surrounding stromal cells. This can be induced by cell intrinsic events (genetic or epigenetic aberrations) or by external factors originating from direct cell–cell contact or from indirect cell communication. While soluble factors, such as chemokines, cytokines, and growth factors, have long been known for their role during tumor development, extracellular vesicles (EV) have recently attracted increasing attention as mediators of intercellular communication. Conserved from prokaryotes to eukaryotes, the secretion of EV is observed in all cell types [2]. The term EV encompasses all types of secreted membrane-enclosed vesicles, which are highly heterogeneous. On the basis of various characteristics, ranging from size, biogenesis, cell of origin, morphology, and content, EV are categorized into four main classes: endosomal-derived small exosomes (Exo) (50–150 nm), plasma membrane-derived middle-sized microvesicles (MV) (100–1000 nm), and large oncosomes (LO) (1000–10,000 nm), as well as apoptotic bodies (500–4000 nm) that are released from dying cells [3,4]. Although commonly used to categorize EV, this classification has been challenged by recent evidence demonstrating that, for instance, the Exo contain further subtypes with different biological and biochemical properties [5]. In order to avoid terminological ambiguities, the term “small EV” was coined for EV harvested at > 100,000 g. However, since most authors still use the term Exo, we will adhere to this nomination throughout the review.

In cancer, EV have been shown to be essential for various steps during tumor initiation and progression. By horizontal transfer of bioactive molecules between cancer cells and the neighboring stroma, EV interfere with signaling and regulation of gene expression in the recipient cell. Thus, the malignant cells can convert the phenotype of surrounding benign cells into a tumor-supporting one and create a favorable environment for cancer progression and metastatic spread. While much attention has been paid to the role of Exo in cancer, the function of the larger MV is still poorly defined. This seems surprising, since MV, in contrast to Exo, are easily accessible in patients’ blood and are characterizable by routine methods that should make them ideal candidates for “liquid biopsies”. Additionally, MV have long been known for their involvement in metastasis formation. This was initially attributed to their procoagulant activity, favoring the formation of microthrombi and facilitating the extravasation of the thus captured circulating tumor cells (reviewed in [6]). However, more recently, accumulating evidence points to a plethora of different ways in which MV are involved in the various steps of tumor progression.

In this review, we discuss the current knowledge about the role of MV during cancer development and shed light on how these insights can be exploited clinically for cancer diagnostics. Moreover, we highlight the divergent features of MV compared to Exo as cancer therapeutics and illustrate the various ways in which MV can either promote or counteract cancer therapy. For this purpose, we specifically included studies in which EV were harvested at 10,000–20,000 g and collectively refer to them as MV.

## 2. Preparation of MV

In order to decipher the role of MV in cancer development and progression, effective methods are required that allow for the stringent isolation of pure MV populations from different cell types and biological fluids. A major caveat in MV research is that the currently available isolation methods potentially co-isolate LO or Exo, yielding a mixed population of EV. This may explain many of the apparently conflicting results in the field of EV research. To address this major challenge, new technologies are under development, but they are not yet suitable for laboratory use. In an endeavor to standardize the experimental procedures and limit experimental variability in the field, scientists of the International Society of Extracellular Vesicles (ISEV) published a position paper indicating the appropriate methods for isolation of EV from cells or biological fluids and highlighting the current knowledge and major caveats of these procedures [7]. A variety of methods are available on the basis of different principles for enriching the various EV subpopulations, including density gradient centrifugation, size-exclusion chromatography, precipitation via volume-excluding polymers, immunoaffinity capture methods, high-pressure liquid chromatography, field flow fractionation, and flow cytometry [7,8]. 

To date, differential centrifugation is still the method of choice for isolating MV since it yields a reasonably good separation of MV from Exo with regard to protein and RNA content as well as function [9,10,11]. In principle, samples are spun down at 2000 g to initially pellet large EV such as LO, followed by centrifugation at 10,000–20,000 g to sediment the middle-sized MV, while ultracentrifugation at ≥100,000 g leads to a harvest of small Exo [8]. Although ultracentrifugation has been criticized for inducing vesicle aggregation and thus affecting downstream applications [12,13,14], these studies were conducted on the 100,000 g Exo pellet. It is unclear to what extent these problems also apply to the 10,000 g MV pellet as well. Another popular method to separate EV from contaminations with non-vesicular proteins or RNA aggregates is ultracentrifugation of EV samples on density gradients. However, the typically used sucrose gradients lack sufficient resolution to separate EV that have slightly different densities and are released by different mechanisms [15]. Since the density of MV and Exo is comparable, they cannot be separated on sucrose gradients easily [11]. Likewise, precipitation methods such as protein organic solvent precipitation (PROSPR) have been demonstrated to co-isolate MV, but the preparations were mainly enriched in smaller Exo, with MV being only a minor side population [16]. Immunoaffinity capture methods rely on antibodies for specific EV surface markers that are coupled to beads and used to isolate specific vesicle subpopulations. Due to the current lack of specific markers for MV, this method has been scarcely used for the preparation of MV. Currently, novel micro-/nano-based devices for MV isolation from clinical samples are being developed (reviewed in [17]). However, one major problem is that the thus isolated vesicle preparations have often been poorly characterized and it is unclear whether they yield MV with a purity comparable to sequential centrifugation protocols. The same applies to methods originally established for Exo isolation such as precipitation or size-exclusion chromatography. While none of the currently available isolation procedures for MV and Exo yield pure vesicle populations, but rather MV- or Exo-enriched fractions, more effective techniques that separate pure MV from Exo might help to shed light on their distinct cell biological features such as biogenesis, uptake, and cargo trafficking routes. Then again, from a clinical viewpoint, the isolation of pure MV populations might not be absolutely necessary for their use as biomarkers, since recent studies have demonstrated the potential of MV for cancer diagnostics or therapy monitoring, even when the preparations contain some smaller Exo.

No matter which method is chosen for EV isolation, the researchers of the ISEV community highly recommend validating the obtained EV by different techniques [7]. This includes the analysis of marker protein expression comprising (i) transmembrane, (GPI)-anchored, as well as cytosolic EV marker proteins (e.g., CD63, CD9, Alix, Syntenin, Rgap1, along with negative controls consisting of (ii) non-EV co-isolated proteins (e.g., ApoB, albumin), (iii) proteins typically present in non-EV subcellular structures such as the Golgi or endoplasmatic reticulum (e.g., GM130, Calreticulin, histones), and (iv) secreted proteins (e.g., collagen, epidermal growth factor (EGF), interleukins). Moreover, EV should be characterized by at least two distinct techniques including their visualization by, for instance, electron or atomic force microscopy, as well as the analysis of their biophysical properties via, for example, nanoparticle tracking analysis (NTA) or Raman spectroscopy. Nanoparticle tracking analysis (NTA) is the most suitable method for analyzing the isolated MV in terms of quantification and sizing. NTA tracks the Brownian movement of laser-illuminated particles and calculates the diameter based on the Stokes–Einstein equation [18]. While NTA remains the most commonly used method for quantitative MV analysis, other methods such as tunable resistive pulse sensing or dynamic light scattering are also available [19]. However, a major limitation of these methods is that they cannot efficiently analyze larger vesicles and that they do not yield any information on the molecular composition of the MV. They are, therefore, combined with other methods such as tunable resistive pulse sensing and Raman spectroscopy or NTA with fluorescence labelling to obtain more information about the isolated MV [19].

A major caveat in MV research is that the currently available isolation methods potentially co-isolate LO or Exo, yielding a mixed population of EV. This may explain many of the apparently conflicting results in the field of EV research. To address this major challenge, new technologies are under development, but are not yet suitable for laboratory use. In an endeavor to standardize the experimental procedures and limit experimental variability in the field, scientists of the International Society of Extracellular Vesicles (ISEV) published a position paper that indicated the appropriate methods for isolation of EV from cells or biological fluids and highlighted the current knowledge and major caveats of these procedures [7]. Furthermore, the researchers of the ISEV community highly recommend validating different techniques for various cell types and biological fluids. A crowdsourcing knowledge base was established to create further transparency with regards to experimental and methodological parameters of EV isolation (http://evtrack.org) [20]. This platform encourages researchers to upload published and unpublished experiments, thereby creating an informed dialogue among researchers about relevant experimental parameters. This represents a major step in facilitating standardization in EV, as well as MV, research.

## 3. Biogenesis of MV

MV directly bud off from the outer cell membrane. The shedding process comprises molecular rearrangements of the plasma membrane regarding lipid and protein composition as well as Ca^2+^ levels. Ca^2+^-dependent aminophospholipid translocases, flippases, floppases, scramblases, and calpain drive the translocation of phosphatidylserine from the inner to the outer membrane leaflet, which is considered a typical characteristic of MV (comprehensively reviewed in [21]). Apoptotic bodies, which are larger in size, also externalize phosphatidylserine on their surface [22,23]. Therefore, the isolation of MV should be conducted solely from healthy and viable cells to avoid contamination with apoptotic bodies, which otherwise can be difficult to discriminate from MV. Ca^2+^ levels regulate membrane rigidity and curvature and maintain physical bending of the membrane, which leads to restructuring and contraction of the underlying actin cytoskeleton enabling MV formation and pinching (reviewed in [24]). MV formation and release are also affected by the lipids ceramide and cholesterol [25]. Neutral sphingomyelinase activity, which hydrolyses lipid sphingomyelin into phosphorylcholine and ceramide, was shown to be involved in Exo and MV release. Inhibition of the enzyme led to a reduction in Exo release, while simultaneously increasing MV budding [26], which suggests that the release of both EV subpopulations is interconnected, albeit on the basis of distinct biogenetic mechanisms.

In addition to lipids, enzyme machineries involved in cytoskeletal regulation play a key role in MV formation and budding. One example is the small GTPase protein ADP-ribosylation factor 6 (ARF6), which stimulates phospholipase D (PLD) that subsequently associates with extracellular signal-regulated kinase (ERK) at the plasma membrane. ERK activates a signaling cascade downstream of the myosin light chain kinase (MLCK) that results in contraction of actinomyosin and enables MV release [27]. Similar to MV, LO are thought to derive from the plasma membrane. However, in contrast to MV, the shedding of LO has been exclusively attributed to aggressive cancer cells that have acquired an amoeboid phenotype to facilitate motility and invasiveness ([4]). Other Rho family small GTPases such as RhoA and RHO-associated protein kinases are equally important regulators of actin dynamics relevant for MV formation [28]. In addition, the endosomal sorting complex required for transport (ESCRT), which is mainly known for its role in the biogenesis of Exo [29], is also involved in MV formation and the last phase of their release. Interaction of arrestin domain-containing protein 1 (ARRDC1) with the late endosomal protein tumor susceptibility gene 101 (TSG101) results in relocation of TSG101 from the endosomal to the plasma membrane, which then induces the release of MV [30].

## 4. Membrane Composition of MV 

EV-mediated cell–cell communication requires targeting and uptake into the recipient cell to deliver the bioactive cargo, which then induces functional and phenotypical changes. These events depend on the composition of the EV membrane, as surface molecules on EV are responsible for binding and docking to recipient cells [31,32]. The molecular composition of the MV membrane closely resembles that of the parental cell [33]. It is enriched in phospholipid lysophosphatidylcholine, sphingolipid sphingomyelins, acylcarnitine, and fatty acyl esters of l-carnitine [34]. ARF6, a key regulator of MV biogenesis, was shown to mediate MV surface molecule selection by recruiting proteins such as ß1 integrin receptor, major histocompatibility complex (MHC) class I and II molecules, membrane type 1-matrix metalloproteinase (MT1-MMP), vesicular SNARE (v-SNARE), and vesicle-associated membrane protein 3 (VAMP3) to tumor MV [27]. Moreover, the bioactive cargo of MV depends on the conditions the parental cells are subjected to, such as inflammation or other stressors. An additional example is hypoxia, which induces recruitment of the RAS-related protein Rab22a to the site of MV budding in breast cancer cells, thus influencing MV formation and loading [35]. 

Since both LO and MV are derived from the plasma membrane, it is not surprising that some transmembrane proteins are present on either of these EV. Analysis of protein marker expression on LO and MV revealed a common signature, underlining the fact that the definition of MV-specific markers remains challenging. Of note, some of the markers initially thought to be specific for Exo, including tetraspanins (CD9, CD63, CD81, HSP60, HSP70, HSP90), membrane transporters and fusion proteins (annexin, flotillin), and multivesicular body (MVB) synthesis proteins (Alix, TSG101) were also found in varying amounts on MV and LO [9]. The fact that, despite their different routes of origin, EV share some common surface molecules and MV-specific markers are still lacking represents major challenges in EV research. It further emphasizes that to correctly characterize EV populations it is indispensable to combine a variety of parameters in addition to marker expression, such as size, sedimentation coefficient, and others. 

## 5. Role of Cancer-Associated MV and Their Protein Cargo in Tumor Progression

EV-mediated cell–cell communication within the tumor microenvironment (TME) is highly complex and has recently been comprehensively reviewed in [36,37]. The role of EV in promoting tumor progression has mostly been elucidated in studies on mixed populations of EV without focusing on a specific subpopulation. Therefore, the specific contribution of MV in this complex process remained enigmatic for a long time. However, with increasing awareness of the presence of large MV, accumulating evidence has begun to unravel the tumor-promoting role of MV in TME communication, as described below by selected examples. The function of MV largely depends on their bioactive cargo, in particular the shuttling of tumor-specific proteins to the surrounding cells. While researchers initially concentrated on the role of nucleic acids transported via EV (e.g., DNA, mRNA, or miRNA), the focus has more recently shifted towards the analysis of the EV proteome. The protein content of MV has been found to be strikingly different from that of the Exo proteome and is enriched in proteins involved in microtubule/cortical actin and cytoskeleton networks, ARF6, its effector phospholipase D2, and parts of the ESCRT family (ESCRT-I) [27,38]. Carrying these bioactive cargo molecules, MV are able to influence either the adjacent tumor cells in an autologous way or the neighboring stromal cells in a heterologous kind of cell–cell communication. 

MV-mediated autologous communication transfers oncogenic traits between tumor cells, resulting in enhanced tumor growth and progression. One example is multiple myeloma in which MV shed by the cancer cells were shown to enhance tumor cell proliferation and thereby stimulate tumor growth [39]. Interestingly, this effect was related to the enrichment of the extracellular matrix metalloproteinase inducer (EMMPRIN/CD147) on the tumor MV. This protein is known to be frequently overexpressed in solid tumors as well as in some lymphomas and leukemias [40]. In line with this finding, another study in breast cancer cells showed that the highly glycosylated isoform of EMMPRIN in particular is present in high levels on breast cancer cell-derived MV and stimulates tumor cell invasion via activation of p38/MAPK signaling [11]. Strikingly, a similar, high-EMMPRIN expression was also found on blood MV from patients with metastatic breast cancer where it was co-expressed with the tumor marker Mucin-1 (MUC1/CA 15-3) [11]. Another tumor-specific factor implicated in pro-tumorigenic tumor–tumor crosstalk via MV is the truncated oncogenic form of the epidermal growth factor receptor (EGFR), EGFRvIII, which is commonly expressed in aggressive brain tumor cells. It was shown that MV secreted by U373 glioma cells contained EGFRvIII, enabling them to transfer malignant characteristics from highly aggressive tumor cells to EGFRvIII-negative, more benign tumor cells, thereby promoting their oncogenic transformation [41]. Hence, MV are convenient communicators within the TME that can either mediate the horizontal transfer of oncogenic material or activate oncogenic signaling pathways in neighboring cancer cells, stimulating their proliferative, survival, mitogenic, and angiogenic potential and shifting them to a highly invasive phenotype.

In addition to tumor–tumor communication, MV were also found to mediate reciprocal crosstalk between the tumor and the surrounding stroma cells. Such heterologous interactions were observed, for instance, between tumor cells and surrounding immune cells, being seemingly essential for cancer immune evasion. As shown for breast cancer cells, the secretion of both tumor MV as well as Exo induced the expression of Wnt5a in tumor-associated macrophages. Macrophage Wnt5a was then, in turn, delivered to breast cancer cells via macrophage-derived MV and Exo, where it activated ß-catenin-independent Wnt signaling, leading to increased tumor invasion [42]. This example indicates that EV-based cell–cell communication can occur bidirectionally in a reciprocal loop and reprogram tumor-associated immune cells towards a tumor-supporting phenotype. The finding that MV-enriched preparations isolated at 50,000 g induced the differentiation of monocytes producing the anti-inflammatory cytokine IL-10 further supports this notion [43,44]. In line with this, early stimulation with tumor MV triggered macrophage polarization towards an anti-inflammatory phenotype with decreased anti-tumor cytotoxic potential [45]. 

Apart from macrophages, other immune cells can be affected by MV, such as the T cells. As T cells represent the first line of the immune defense, tumor cells seem to have established strategies to suppress T cell activity and dampen antitumoral immune response by exploiting MV-mediated cell–cell communication. For instance, leukemia-derived MV deliver miRNAs to T cells, which then interact with their targets, resulting in a T cell exhaustion phenotype [46]. Moreover, MV released by irradiated breast cancer cells were shown to carry an increased amount of immune-modulating proteins, such as programmed cell death ligand 1 (PD-L1). Via transfer of this immunosuppressive protein, tumor MV inhibited cytotoxic T cell activity and enabled tumor growth [47]. 

Tumor cell-derived MV were also shown to modulate fibroblasts in the tumor stroma. Stimulation of fibroblasts with prostate cancer MV converted them to an activated phenotype and triggered the release of tumor-promoting fibroblast MV [48]. Similar observations were made in oral squamous carcinoma when normal human fibroblasts were treated with tumor-derived MV [49]. In this study, the switch to cancer-associated fibroblasts (CAF) was mainly mediated via metabolic reprogramming of the fibroblasts to aerobic glycolysis, with an increase in glucose uptake and lactate secretion. Co-culturing the generated CAF with oral squamous cell carcinoma cells again led to enhanced cancer cell invasion and migration. Interestingly, MV-induced fibroblast activation and spreading seems to occur in particular in the stiff matrix environment that is typically found in the tumor periphery [50]. Tumor MV are also able to modulate angiogenesis within the primary tumor. In a study on a mixed population of MV and Exo, normal endothelial cells were found to endocytose tumor EV, which activated PI3K/Akt signaling and promoted the motility as well as the tube formation activity of the endothelial cells [51]. As the pro-angiogenic factor vascular endothelial growth factor (VEGF) has been found on MV and Exo, this might be one important factor contributing to the stimulatory effect of tumor MV on endothelial cells [52]. Of note, MV transport a unique 90 kDa form of VEGF that has only a weak affinity for bevacizumab, the clinically used monoclonal anti-VEGF antibody, which might thus be ineffective in blocking MV-mediated activation of the VEGF receptor (VEGFR) [53]. Similarly, MV derived from multiple myeloma cells were shown to transfer CD138, a myeloma cell marker, to endothelial cells. This stimulated the endothelial cells to proliferate, invade, and secrete the angiogenic factors IL-6 and VEGF to promote tube formation [54]. Taken together, these observations emphasize the important role of MV in tumor–stroma crosstalk during tumor progression. 

Using the same mechanisms as discussed above, MV were found to modulate the surrounding environment beyond the primary tumor by shaping the formation of pre-metastatic niches over long distances. In pancreatic cancer, tumor MV were able to enter the liver microcirculation and extravasate through the vessel wall in a CD36-dependent manner where they were taken up by perivascular macrophages and primed the liver metastatic niche [55]. Furthermore, the highly metastatic melanoma cell line B16F10 releases large amounts of tumor MV into its surroundings. These MV were found to be able to induce metastasis formation in BALB/c mice, which are normally resistant to the B1610 tumor cell line [56]. Metastatic niche formation might be essentially influenced by tumor MV uptake into organ-specific macrophage populations, and this uptake seems to be further influenced by systemic changes such as inflammation [57]. However, further in vivo studies are required to characterize the uptake kinetics and to clarify which cells are specifically targeted by tumor MV during systemic spreading. In summary, an increasing number of studies confirms that MV play a critical role in cell–cell communication within the TME and support local tumor growth and progression as well as tumor spreading to distant sites through the priming of metastatic niches (Figure 1). 

## 6. Analysis of MV in Peripheral Blood

MV have been successfully detected in almost all body fluids, including saliva [58], urine [59], cerebrospinal fluid [60], breast milk [61], ejaculate [62], synovial fluid [63], bronchoalveolar lavages [64], and blood, wherein they were described as early as 1967 by Peter Wolf as “platelet dust” [65]. The advantage of body fluids apart from blood is that they contain MV specifically enriched for the cancers drained by them. Nevertheless, some of these liquids are either difficult to obtain or provide additional methodological problems. For instance, MV in urine are of special interest as biomarkers for urothelial cancers. However, their isolation poses two additional challenges: (1) the most abundant protein in urine, the Tamm–Horsfall protein, associates and co-precipitates with EV, and (2) standard protocols pre-clear urine samples at 17,000 g to deplete the urinary sediment, and thus also deplete the MV contained within. Therefore, the number of studies on MV in urine meeting the criteria mentioned in the introduction is very limited. Since blood is the most thoroughly studied body fluid, as well as the most likely application route for using MV with regard to diagnostic and therapeutic interventions, we chose to focus this review on MV in blood. 

When isolating MV from blood, several studies have shown that the choice of anticoagulant used for blood collection has a critical influence on the obtained MV, although the results seem to be contradictory. While the use of protease inhibitors, either sodium heparin or a mix of hirudin supplemented with a factor Xa inhibitor, was initially recommended because of the high number of preserved MV [66], other reports have argued for the use of sodium citrate since the lower numbers of MV might point to less artificial MV shedding [67]. In contrast, Jamaly et al. recently used various anticoagulants to isolate MV and measure their total plasma concentrations by NTA without finding any significant differences in numbers [68]. Apart from total MV numbers, the isolation of MV from serum drawn into lithium heparin tubes seems to result in the aggregation of platelet-derived MV, with sticky vesicle pellets and a reduction in platelet MV numbers, thereby specifically affecting certain MV subpopulations [69,70]. In addition to the anticoagulant, other preanalytical factors such as centrifugation, temperature, freeze–thaw cycles, or agitation can influence MV in blood samples, which should be taken into account when setting up isolation protocols for clinical studies [66,67,71]. However, long-term storage of plasma samples without repeated freeze–thaw cycles seems feasible without significant MV loss [67,69,72]. Another important factor for the analysis of large MV in blood is the initial removal of platelets and blood cells to obtain platelet-poor plasma (PPP). Since PPP is often prepared by centrifugation at 2500–3000 × *g*, this step might result in a significant loss of larger MV, which already pellet at this force [9]. It has been proposed that two centrifugation steps at 1500× *g* might be preferable for MV isolation [70]; however, more elaborate sorting methods might be required to specifically isolate larger EV from blood. 

A current study by Brennan et al. compared different EV isolation methods, including ultracentrifugation, density gradient centrifugation, and size exclusion chromatography, as well as different combinations thereof, for their potential to isolate EV from plasma [73]. While none were found to be clearly superior, the authors showed that the isolation method significantly affected the yield, size, and degree of contamination with serum (lipo)proteins [73]. This must therefore be considered carefully on the basis of the planned downstream application. Unfortunately, the study was conducted on whole EV preparations, and thus it is again unclear whether the results are also applicable for larger MV. A novel approach to specifically reduce the contamination of plasma EV preparations with lipoproteins is to use anti-apolipoprotein B (ApoB) antibody-coated magnetic beads prior to EV isolation [74]. However, the method might lead to a high loss of vesicles. 

The preferred method for analyzing MV from clinical samples is flow cytometry because of its wide availability in most labs and clinical centers, its capability for rapid preparation and acquisition of a large number of samples, as well as its potential for standardization. Since the size threshold of standard flow cytometers is ~300 nm (see [75] for a current instrument comparison), this allows the quick measurement and quantification of MV protein expression in contrast to Exo, which have to be coupled prior to analysis to larger latex beads following a time-consuming protocol. Protocols exist for staining of single or multiple markers on plasma-derived MV for flow cytometry [76,77]. Even though flow cytometry might not allow the detection of very small MV <300 nm and thus not reproduce the whole spectrum of MV in blood, several studies have successfully demonstrated the potential of flow cytometry for measuring MV-associated cancer biomarkers, as discussed in detail below. However, analysis of MV samples by flow cytometry bears several pitfalls, and the settings should be chosen with care to avoid misinterpretation of the results [78]. When measuring MV by flow cytometry, their presence should be confirmed by additional methods such as NTA or electron microscopy as the correct sizing of vesicles by flow cytometry can be challenging [79]. When discriminating MV from background noise using annexin V staining, the use of saline was recommended since the routinely used PBS seemed to create artificial nano-sized vesicles and could thus lead to false positive results [80]. Another pitfall is that the insoluble fraction of immune complexes and MV have overlapping size profiles, which might interfere with their flow cytometric analysis. To confirm that MV and not immune complexes are measured, samples can be treated with low concentrations of detergents (e.g., 0.05% Triton X-100), which lyse MV, thus resulting in a loss of signal in contrast to immune complexes that keep their fluorescent signal [63]. Recently, another label-free approach called flow cytometry scatter ratio (Flow-SR) has been introduced to discriminate MV <500 nm from lipoprotein particles, which are highly abundant in plasma samples [67].

Apart from flow cytometry, there are protocols for surface profiling of plasma MV by antibody microarrays. However, this is a semiquantitative method and thus more useful for initial screening of patient samples for the expression of potential novel biomarkers [81]. In summary, the current studies have shown that isolation and analysis of MV from human biofluids, such as plasma, is feasible using standard laboratory equipment. However, it is clear that transparent reporting and standardization of protocols are highly important for implementing and translating MV as novel biomarkers and therapeutics into the clinic [82,83]. 

## 7. MV Levels in Cancer Patients

Most studies have focused on measuring total MV levels in the plasma of cancer patients. In this context, it must be kept in mind that tumors can secrete large numbers of apoptotic bodies under therapy due to the massive induction of apoptosis in the tumor tissue. Apoptotic bodies can have a similar size as MV and are equally characterized by an externalization of phosphatidylserine, a marker that is often used to identify MV by flow cytometry. Hence, care should be taken to include only patients prior to treatment into these studies. While the observed results are very contradictory, the majority of reports have documented increased MV levels in cancer patients compared with healthy controls. Augmented MV levels have been detected by flow cytometric quantification, mostly using TruCount Beads, in the plasma of patients suffering from gastric cancer [84], lung cancer [85], breast and pancreatic cancer [86,87], and colorectal cancer [88], as well as in most hematologic malignancies including chronic lymphocytic leukemia (CLL) and multiple myeloma [89,90,91]. The same trend has been confirmed for hepatocellular carcinoma by measuring MV protein levels by bicinchoninic acid assay (BCA) [92]. In contrast, two studies investigating heterogeneous cohorts of patients with different types of advanced cancers did not detect significant differences in total plasma MV numbers by flow cytometric quantification or analysis of MV protein content [69,93]. Similar observations were made by a novel capture/imaging approach in glioblastoma patients [94]. Another report quantified the total number of annexin V+ MV by flow cytometry and found a significant decrease of MV levels in patients with colorectal cancer and benign colorectal diseases compared with healthy controls [95]. The discrepancy between the observations might be due to the methods used for MV quantification, the composition of the patient and control cohorts, or diverging effects in different cancer subtypes. Patient selection criteria and the clinical setting of sample acquisition (before/during/after treatment) are not always conclusively reported in the studies, which could be another reason for the heterogeneous results. 

While the increased levels of MV in hematologic malignancies could be explained by additional vesicle shedding from the large numbers of cancer cells in blood, the source of the elevated MV counts in solid tumor patients remains undefined. In these patients, plasma levels of tumor MV might depend on the access of the tumor to the circulation or might originate from other blood cell populations as a reaction to the growing tumor. Other limitations are that all the studies are either single center studies; include a very small number of patients; or have compared MV levels between cancer patients and younger, healthy controls. It was recently found that total MV levels were higher in healthy, untreated elderly than in younger individuals, which might have biased the results [96]. 

An argument for a specific increase in MV levels in cancer patients is that MV levels seem to increase with advanced tumor stages and have been found to be elevated in late stage metastatic breast cancer [86,87], colorectal cancer [88], hepatocellular carcinoma [92], or CLL [90] compared with early stage patients. Of note, a significant increase was already observed in stage I colorectal cancer patients compared with healthy controls [88]. In line with this, total MV levels decreased 1 month after surgery in hepatocellular carcinoma patients and have thus been suggested as a potential diagnostic biomarker [92]. Moreover, higher MV levels in blood might additionally serve as a prognostic biomarker since they were found to correlate with a poor clinical outcome in glioblastoma or CLL [90,97]. In contrast, in lung cancer, an increase in MV in blood was associated with better progression-free survival [85]. An interesting finding was the detection of very large vesicles (1–14 µm), measurable with the established CellSearch system used for the analysis of circulating tumor cells (CTCs) [98]. These epithelial cell adhesion molecule (EpCAM)+/cytokeratin CK+/CD45- large vesicles without nuclei were present at frequencies one order of magnitude higher than CTCs and were elevated in the plasma of prostate, breast, and colorectal, but not lung cancer patients, with high numbers correlating with shorter overall survival [99]. While these vesicles have not yet been validated to be “true EV” using the MISEV guidelines, their discovery might represent one additional opportunity of translating EV as biomarkers into the clinic using an instrument already applied in diagnostics. 

## 8. MV as Cancer Biomarkers

The growing number of in vitro studies demonstrating that cancer cells shed large numbers of EV with tumor-specific material has fueled interest in using EV as biomarkers in the clinic. Most investigations thus far have focused on the detection of tumor-derived vesicles within the complex mixture of plasma EV. This seems to be a promising strategy for gaining information about the genetic makeup of the tumor and identifying targetable characteristics, in particular for solid tumors, which are not directly accessible without invasive procedures (e.g., pancreatic cancer, glioblastoma). In contrast to the classical soluble “tumor markers” in the serum, MV also allow for detection of non-secreted, membrane-bound, or even intracellular molecules. Thus, the MV-based liquid biopsy enlarges the spectrum of detectable tumor-associated factors that could help not only with regard to diagnosis but also with accurate patient risk stratification and therapy monitoring. Since tumors seem to gain access to the circulation quite early [100], tumor MV in blood are believed to be of potential interest as early diagnostic biomarkers. In contrast to the extremely sparse CTCs, they are present in significantly greater numbers, which might improve detection sensitivity. Recently, tumor EV, consisting of a mixture of small and large vesicles, were shown to be able to pass the intact blood–brain barrier in an orthotopic glioblastoma xenotransplant model of human cancer stem cells as well as in glioma patients [101]. In line with this finding, EV carrying tumor-specific alterations were detected in the blood of patients with low-grade glioma, most of whom still had an intact blood–brain barrier [101], thus further supporting the concept of using EV as early cancer biomarkers. Another important point is that tumors are known to be highly heterogeneous [102], a circumstance that currently gains more and more attention with the novel single cell analysis technologies. While a biopsy taken from a limited area within the tumor only partly mirrors the molecular composition of the whole tumor, EV are believed to give a more comprehensive picture. This might apply also to patients with advanced, metastatic cancer, considering that metastases often differ from the primary tumor, as well as among each other, and might thus express different markers or potential therapeutic targets [103,104]. Interestingly, an early study in glioblastoma described the detection of oncogenic EGFRvIII transcripts in plasma-derived EV in two patients who had been diagnosed as EGFRvIII-negative on the basis of a tumor tissue biopsy [105]. Thus, it may be speculated that EV might mirror tumor heterogeneity more accurately than limited biopsies. 

When aiming at the specific detection of tumor-derived MV in blood, the first step is to identify cancer-specific factors that are not expressed on MV shed by other cell populations. Therefore, a thorough characterization of tumor MV is necessary, which usually has been performed on cancer cell line-derived vesicles for later translation to patient samples. A recent report has suggested that MV reflect the cellular transcriptomic landscape better than Exo and can thus give valuable information about mutation status or gene amplifications that can be used for cancer diagnostics [106]. In line with this, hallmark oncogenic fusion transcripts (e.g., BCR-ABL1, TEL-AML1, MLL-AF6) as well as other specific tumorigenic transcripts (e.g., HOXA9, MEIS1) were found in MV of leukemia cell lines [107,108]. Similarly, in a glioblastoma model, genomic DNA (gDNA) sequences of tumor-associated genes were identified with a distinct distribution in either apoptotic bodies, MV or Exo isolated from human cancer stem cells, or from the blood of xenotransplanted mice, including specific detection of PIK3CA, EGFR, AKT1, and MDM2 sequences in MV [101]. However, the question remains as to whether analysis of vesicle-associated transcripts is superior to mutation detection based on cell-free DNA. Encouraging results have been obtained in this context by the group of Krug et al., who demonstrated increased sensitivity in detecting EGFR mutations in the plasma of lung cancer patients with a combined analysis of circulating tumor and EV DNA compared to circulating tumor DNA alone [109].

MV and Exo have been shown to differ in their RNA composition [10]. While several population-specific microRNAs have been identified for both MV and Exo, there seems to be a trend that miRNAs are relatively more enriched in Exo and mRNAs in MV [106,110,111]. In addition, an enrichment compared with the whole cell has been observed as several miRNAs that were almost undetectable in colorectal cancer cell lysates (<5 transcript per million (TPM)) were found at high levels (>1000 TPM) in the respective MV and Exo from the cells [111]. However, it is currently unclear whether sufficient amounts of RNA molecules are shuttled to EV in order to functionally influence recipient cells, as several critical reports have revealed that most RNA species are present at frequencies of much less than 1 copy per EV [106,110,112]. Therefore, the focus of the rest of this review is on MV-associated proteins that have the additional benefit that they could be used not only as biomarkers, but also for specific drug delivery and targeting. The protein content of EV has been evaluated largely by Western blotting, flow cytometry, as well as different proteomic approaches. While shot-gun proteomics was initially used to thoroughly characterize the whole EV proteome, novel targeted proteomic acquisition strategies such as selected (or multiple) reaction monitoring (SRM/MRM) allow the detection of a predetermined selection of specific target peptides with high sensitivity and quantitative accuracy [113]. These methods thus facilitate the discovery and large-scale validation of biomarker pipelines in complex biological samples (e.g., blood) [114]. In a recent study, SRM/MRM was successfully employed to identify novel potential biomarkers for prostate cancer on Exo in urine [115], demonstrating its potential for future EV-related biomarker studies.

The MV proteome has been proposed to be more similar to the cellular proteome than that of Exo [35]. However, not all proteins from the cellular plasma membrane are found on MV, and some are highly enriched, which suggests specific sorting mechanisms [116]. An overview of MV-associated tumor antigens that have been successfully detected in the blood of cancer patients and used as biomarkers is presented in Table 1. In hematologic malignancies, MV carrying the respective malignancy-associated antigen (e.g., CD38 for multiple myeloma, CD30 for Hodgkin lymphoma) were detected in cancer patients but were almost undetectable in healthy controls [91]. Similar observations have been made for solid tumors. In one of the earliest studies, the oncogenic receptor human epidermal growth factor receptor 2 (Her2/Neu) was identified and significantly elevated on blood-derived MV from gastric cancer patients [84]. Meanwhile this finding has been confirmed for colorectal cancer [117]. In the largest study to date of a cohort of 330 cancer patients with advanced solid tumors and 103 healthy, non-cancer controls, the number of MV carrying the tumor antigen EMMPRIN were found to be increased with advanced tumor stage [69]. Expanding the analyses on MV co-expressing EMMPRIN together with other tumor-associated molecules, such as epithelial cell adhesion molecule (EpCAM/CD326), MUC1/CA 15-3, or EGFR1, the numbers were not only significantly elevated in cancer patients versus controls but also associated with poor overall survival, thus suggesting that MV might indeed represent valuable prognostic cancer biomarkers. EpCAM is among the most studied antigens as it is known to be overexpressed on tumors of epithelial origin and is thus a bona fide tumor marker. The combination of EpCAM with other tissue-specific markers such as, for instance, the asialoglycoprotein receptor 1 (ASGPR1), exclusively expressed in liver cells, allowed the separation of patients with liver disorders into patients with or without liver tumors (including hepatocellular or cholangiocellular carcinoma) [118]. In line with this, the EpCAM+/EMMPRIN+ MV levels correlated with tumor size in colorectal cancer patients and decreased after surgical removal of the tumor [118,119]. In yet another study performed on plasma samples from metastatic prostate cancer patients, caveolin-1 (CAV1) was found on 5–10% of large EV (1–10 µm) isolated by filtration, but with a size profile similar to the 10,000 g pellet, whereas it was barely detectable in healthy controls [120]. 

From the thus far published MV biomarker studies, one can deduce that the population of tumor MV carrying the respective tumor antigens constitutes up to 5–10% of the total MV in blood of advanced cancer patients [69,94,120]. A recent study has profiled MV at the single vesicle level and proposed that there is a huge heterogeneity among MV, and that not all MV, even from the same cell line, carry the respective tumor markers [94], which further hampers the detection of tumor MV in blood. Moreover, especially low-abundance tumor MV might be concealed in the heterogeneous mixture of blood MV, which might necessitate the development of novel protocols for specific enrichment of tumor MV populations. One suggestion is to remove the platelet-derived MV prior to analysis [70] since they represent the largest MV population in blood. A protocol was recently published for isolating different subpopulations of small and large EV from solid tumor tissue, in this example from melanoma [122,123]. This is an important step that will now allow for isolation of MV from primary tumor tissues and metastases to define not only tumor-specific but even patient-specific biomarkers. The comparison of MV profiles from the tumor tissues (primary tumor/metastases) with MV present in body fluids might then provide information on clonal evolution and serve as the basis for therapy selection and monitoring.

## 9. MV Biomarker Signatures

The studies conducted thus far have shown that while single MV-associated biomarkers are often elevated in the blood of cancer patients, each of the antigens alone failed to reliably distinguish cancer patients from healthy controls, mainly due to low sensitivity. In contrast, combinations of several markers have proved to be of superior diagnostic value, such as the combination of EMMPRIN/EpCAM/MUC1/EGFR, which reliably separated cancer patients from healthy controls with a high AUC value of 0.85 [69]. It seems likely that more comprehensive signatures comprising cancer subtype-specific markers can even further increase sensitivity and specificity. Importantly, it has to be considered that the overexpression of certain oncogenes might impact MV protein profiles as was shown for breast cancer for the oncogene Src for Exo [124] or Her2/Neu, which was shuttled along with its associated proteins onto MV and Exo [125]. The resulting specific signatures have already been successfully used for the correct determination of breast cancer subtypes from serum EV [126]. Compared to soluble cancer biomarkers, which often suffer from low specificity and a high rate of false positive results, the strength of MV is their potential as biomarker platforms that simultaneously carry multiple markers, allowing for the recognition and detection of specific patterns. 

## 10. The Microvesicle Reactome in Blood

Several conditions, such as hypoxia, inflammation, exercise, or nutrition, influence levels of MV subpopulations in blood (reviewed in [127]). It might thus be worth considering the fact that not only the analysis of tumor MV in blood but also changes in blood MV composition or the markers on distinct blood MV subpopulations might give information about cellular transformation (Figure 2). Although, for example, endothelial cell-derived MV were detected at elevated levels in some hematologic malignancies [91], a slight increase was observed in platelet- as well as leukocyte-derived MV in solid tumor patients [69,88,128]. An analysis of the phospho-proteome of MV and Exo isolated from plasma samples of breast cancer patients by ultracentrifugation described 156 unique phosphosites, which differed significantly between the vesiculome of healthy individuals and patients [129]. More specifically, in the blood of gastric cancer patients, the levels of MV carrying the chemokine receptor CCR6 were elevated. The cell type of origin, however, remained unclear [84]. 

Among the EV subpopulations, MV are considered the main source of tissue factor (TF) activity [130]. Higher amounts of MV with TF activity were detected in the plasma of metastatic cancer patients, suggesting their use as novel markers for venous thromboembolism, a common complication in advanced cancer patients [131]. In addition to TF, phosphatidylserine, whose exposure on the outer membrane is a typical feature of all MV, can further induce thrombin generation [132]. This potential side effect should therefore be carefully taken into account when considering the application of MV as therapeutics, although first studies in mice did not show an effect of the injected MV on coagulation [133].

Taken together, analysis of the whole mixture of plasma vesicles does not only detect the presence or absence of tumor EV but yields more comprehensive information since it additionally mirrors the reaction of immune and other cells against the tumor EV.

## 11. MV for Therapy Monitoring

First studies have revealed the potential of MV for monitoring therapeutic responses in cancer patients. Krishnan et al. reported that plasma levels of CD138+ MV corresponded with therapeutic response in individual multiple myeloma patients [89]. In general, total levels of MV were found to decrease after surgical removal of the tumor, which was generally associated with a better outcome [92,97]. In line with this, the number of EpCAM+ MV decreased 10 days after surgery in colorectal cancer patients, although the effect was not seen for EpCAM+/EMMPRIN+ MV [119]. This might be explained by the additional presence of EMMPRIN on platelet- and immune cell-derived MV [11], which are directly influenced by the surgery. Prolonged high levels of total EV or endothelial cell-derived CD144+ MV after chemotherapy seem to predict a poor clinical response and were found to be associated with shorter progression-free and overall survival [134,135]. The same trend was observed in glioblastoma patients showing tumor progression after radiochemotherapy who generally exhibited higher MV levels compared with patients with pseudoprogression or stable disease [136]. Moreover, an increased concentration of large and small EV prior to therapy was associated with failure of chemotherapy in breast cancer patients [134]. Taken together, these results demonstrate that MV levels not only mirror the presence of tumor, but that MV might also have a protective role against chemotherapeutic agents.

## 12. MV and Cancer Therapy

As discussed above, MV collected by liquid biopsies can give valuable information about targetable characteristics in the growing tumor and thus be used as an innovative read-out for targeted therapy decisions and patient stratification for personalized medicine. However, it has also become clear that MV can interfere with cancer treatment and limit therapeutic success. On the one hand, MV have been identified as transporters for P-glycoprotein (MDR1) and multidrug resistance-associated protein 1 (MRP1), two plasma membrane multidrug efflux transporters, and have thus been recognized as vehicles for spreading multidrug resistance from resistant to sensitive tumor clones [137,138]. On the other hand, cancer cells export cytotoxic agents via their MV during chemotherapy as has been shown, for instance, for doxorubicin, cisplatin, gemcitabine, or docetaxel [133,139,140]. Cancer cells seem to shed higher numbers of MV during treatment, and it has been suggested that the potential of a cell to shed MV correlates with its resistance to treatment [140]. Therefore, it seemed to be a promising strategy to inhibit MV release in order to increase treatment efficiency. Indeed, the combination of chloramine and bisindolylmaleimide-I, two inhibitors blocking the release of both MV and Exo, highly increased apoptosis in cancer cell lines treated with chemotherapeutic agents [141]. In another study, reducing MV shedding by inhibition of calpain via siRNA or the specific inhibitor calpeptin significantly elevated intracellular levels of docetaxel. This increased the susceptibility of the cell to chemotherapy, allowing for treatment with lower doses of chemotherapeutic agent at higher efficiency in vitro as well as in a xenograft mouse model in vivo [139]. Considering the toxic effects of general inhibition of EV secretion, it still remains elusive as to whether this strategy is feasible for human patients or whether it will be possible to specifically target such inhibitors to the tumor tissue. 

Another mechanism as to how EV can impede cancer therapy is that cancer cells have been shown to export therapeutic targets such as CD20, Her2/Neu, or PD-L1 onto Exo [142,143,144] that capture the administered therapeutic antibodies and shield the tumor cells from attack, resulting in therapy failure. Moreover, Exo have been recently shown to transfer resistance-mediating cargo from stromal to cancer cells [145]. Whether the same mechanisms indeed also apply for large MV is currently unknown. 

Considering the evidence for a tumor-supporting role of MV, they should also be regarded as novel targets for therapeutic approaches. This notion is supported by a study by Keklikoglou et al. showing in a mixed population of MV and Exo that chemotherapy induces the release of Annexin A6 (ANXA6+) EV from murine breast cancer cells. When these chemotherapy-induced ANXA6+ EV were injected into the tail vein of mice, they activated endothelial cells to produce the chemokine Ccl2, which led to the expansion of Ly6C+CCR2+ lung monocytes and thus pre-conditioned the lungs for the seeding of breast cancer metastases [146]. Similar observations of EV-mediated pre-metastatic niche formation have been made for melanoma [147]. In line with these findings, MV isolated from cancer patients induced a tumor-supporting phenotype in human macrophages ex vivo and highly increased the invasive potential of benign breast cancer cells, while MV isolated from healthy controls had no such effect [69].

## 13. MV as Cancer Vaccines

Although the studies presented thus far argue for an unfavorable role of MV in hampering cancer therapy, the final two chapters discuss how MV could also be exploited positively with regard to therapeutic interventions. In a first study, Zhang et al. reported that while the immunization of mice with PBS or tumor cell lysates failed to prevent the growth of subsequently injected tumor cells, mice treated with tumor Exo or MV developed tumors in only 87.5% or 50% of the animals, respectively [148]. These initial results suggested that tumor MV, which are enriched in a multitude of tumor-specific proteins (see Table 1), could serve as a novel, cell-free source of tumor antigens that are more immunogenic than other parts of the tumor cells, including tumor Exo. A possible explanation for their superior efficacy compared with Exo might be the presence of DNA as well as mitochondrial fragments inside MV that discriminate them from Exo and might provide an additional immunostimulatory signal for innate immune cells. However, more studies are required to confirm the differential immunogenic effect of MV and Exo and explain the potential difference. 

In general, the anti-tumor immune response is believed to occur largely via cytotoxic CD8+ T cells activated by tumor antigen-presenting dendritic cells. Since immunization with tumor MV prior to tumor growth is hardly possible in human patients, strategies will have to be developed to treat already existing tumors. Indeed, in the above-mentioned study, the injection of tumor MV into mice with already growing tumors failed to induce a sufficient activation of T cells [148]. Instead, the injection of dendritic cells primed with tumor MV overcame the problem of inadequate T cell activation, enhancing T cell infiltration in the tumor tissue as well as tumor cell killing [148]. Mechanistically, incubation of dendritic cells with tumor MV induced dendritic cell maturation with concordant upregulation of the co-stimulatory molecules CD80 and CD86, increasing their homing to the tumor-draining lymph nodes [148,149]. Further evidence that the use of MV as tumor vaccines might indeed lead to successful anti-tumor immune responses has been provided by a recent study by Pineda et al. who harvested MV from C6 rat glioma cells and subsequently administered them to rats, which had been inoculated with a subcutaneous C6 glioma. MV vaccination increased the number of infiltrating T cells in the tumor tissue, induced tumor cell apoptosis, and led to a reduction of tumor growth [150]. Interestingly, it might even be possible to deliver future MV vaccines orally, as it was discovered that tumor MV injected intragastrically into mice are taken up by intestinal epithelial cells in the ileum and transferred by transcytosis to dendritic cells at the basolateral side that can subsequently induce T cell activation and anti-tumor immune responses [151]. However, this treatment was only successful with concordant anti-acidic agents, as MV are sensitive to degradation by gastric acid. 

Thus far, clinical studies testing the efficiency of tumor cell- or Exo-based vaccines have shown only limited efficiency and it remains to be seen whether this trend is different for MV. The efficiency of tumor MV vaccines will critically depend on whether it will be possible, on the one hand, to find ways of boosting their efficacy in eliciting an anti-tumor immune response, while, on the other hand, limiting their immunosuppressive functions. 

## 14. MV as Vehicles for Drug Delivery

The finding that EV transport bioactive molecules (e.g., proteins, metabolites, or nucleic acids) raised interest early on in terms of their use as a novel drug delivery system. Compared with cell-based therapeutics, EV have the advantages that they do not possess the potential for transformation or unlimited growth, they are known to cross biological barriers such as the blood–brain barrier, and as non-living material they can be stored or transported more easily and be modified with more aggressive manipulation techniques. Another positive aspect is their natural origin, which makes them more biocompatible than artificial liposomal drug carriers, with enhanced stability, less immunogenicity, and less liver toxicity [152]. In general, EV as drug carriers offer protection for their content, such as from enzymatic degradation, which enhances cargo stability in biofluids. EV can be modified with several approaches to increase their bioactive potential, including either manipulation strategies for the secreting cell, such as genetic manipulation, cell stress, or hypoxia, or post-isolation loading methods, such as electroporation, sonication, heat-shock, or EV transfection. A current overview of the techniques, their advantages, and their limitations is presented in [153,154]. Although most methods have been solely tested for Exo, the majority should be equally applicable to MV, considering the similarities in their biophysical makeup. 

The first report on MV as a drug delivery system was published by Tang et al., who successfully used chemotherapy-loaded MV to inhibit tumor growth in a murine hepatocarcinoma ascites model, as well as in severe combined immunodeficient (SCID) mice injected with ovarian cancer cells [133]. Delivery was even successful to solid tumors when MV were injected intravenously [133]. In a first in vivo approach, the same group tested the therapeutic potential of intrathoracic injections of cisplatin-loaded tumor MV in six end-stage, cisplatin-resistant lung cancer patients. The injections greatly reduced the number of tumor cells (>95%) in the metastatic malignant pleural effusions in three of the six patients, suggesting an efficacy of the MV injections and also their ability to reverse drug resistance [155]. However, the treatment did not show any benefit in the other three patients, raising the question about the underlying reasons for the treatment failure. Two clinical trials are currently registered to further explore the potential of MV as therapeutic tool in cancer: a phase I/II study investigating the use of red blood cell-derived MV loaded with methotrexate in the treatment of malignant ascites (clinicaltrials.gov identifier: NCT03230708), and a phase II study aimed at generating methotrexate-loaded autologous tumor MV from malignant effusion and testing their effect on tumor growth and immune regulation (clinicaltrials.gov identifier: NCT02657460). 

The majority of the conducted studies thus far has used EV from mesenchymal stem cells as native vehicles for therapeutic applications, and none of them has observed significant side effects [156]. Alternatively, red blood cell-derived EV have been suggested especially for the delivery of RNA drugs since they are easily available, do not contain DNA, and can be easily modified by electroporation [157]. However, in mouse models, their systemic administration as drug carrier was found to only be feasible for the therapy of leukemias, while successful delivery to solid tumors required intratumoral injections due to the otherwise low target organ specificity [157]. In general, EV biodistribution seems to depend on the kind of cell used for EV production, the EV delivery route, the dosage, and the uptake efficiency [158]. After systemic injection, a major part of the injected EV is trapped in the liver, spleen, or lung [158,159], which on the one hand might limit EV distribution to other organs, but on the other hand might be used for specific cargo delivery to these organs. Several engineering strategies are currently being tested to improve EV circulation kinetics and increase targeting to specific cell populations [160]. A hallmark of MV is that phosphatidylserine is exposed on their outer membrane surface, a feature which distinguishes them from Exo. Since externalized phosphatidylserine is a recognition signal for macrophages and triggers phagocytosis [161], MV might be superior for drug delivery to this specific cell population. 

Interestingly, a recent report suggested that the choice of MV or Exo as drug carrier influences the intracellular delivery route of the transported cargo. While MV-loaded paclitaxel was mostly delivered by membrane fusion and endocytosis, Exo-loaded paclitaxel was predominantly taken up by endocytosis into prostate cancer cells [162]. Moreover, MV seem to promote drug entry into the nucleus, even into highly resistant tumor-repopulating cells [155]. Since the mechanism of internalization and intracellular trafficking significantly affects the functional efficiency of the delivered drug, further research aiming at elucidating the different fates of MV- and Exo-delivered cargo will help to choose the most accurate EV delivery system for a given drug. 

While MV are larger and therefore allow for packaging larger amounts of drug molecules, it could be more difficult for them to penetrate into the tissue due to their size. One solution that has been presented suggests isolating MV from tumor cells grown in and adapted to soft fibrin gels, which resulted in the shedding of less stiff MV [163]. These softer MV with a higher capacity for deformation showed enhanced extravasation into the tumor with deep tissue penetration and, as a result, enhanced treatment efficacy when loaded with the chemotherapeutic agent doxorubicin [163]. Moreover, a weaker extravasation was observed in non-tumor tissue compared with MV isolated from cells cultured in 2D [163], suggesting less toxicity. A problem for both MV and Exo seems to be their limited lifespan in circulation. For fluorescently labeled pancreatic tumor MV injected into healthy mice, the signal was cleared after 60 min, while the majority of MV were already lost within 15 min after injection [164]. 

Taken together, while both Exo and large MV have been shown to possess the potential for drug delivery, more comparative studies are required to evaluate common as well as diverging features of both EV populations as therapeutic vehicles. Both still suffer from the lack of scalable, specific, cost-effective production and isolation methods as well as poor drug loading efficiency, problems that need to be solved for their standardized use in the clinic. Moreover, heterogeneity and batch-to-batch variability are known to strongly influence their therapeutic efficacy [165]. 

## 15. Concluding Remarks and Future Perspectives

Over the past decade, an increasing number of reports has indicated that plasma membrane-derived MV are a distinct vesicle population with some common but also several diverging features compared with the smaller Exo. While most researchers have focused on the analysis of Exo, MV offer several advantages as diagnostic and therapeutic tools. In particular, their isolation is easier and less time-consuming and does not necessarily require ultracentrifugation with its associated problems of vesicle aggregation, damage and loss, and instrument availability. Additionally, protocols are available for their rapid and thorough characterization by standard flow cytometry, a technique that is already well-established in routine clinical diagnostics. Interesting findings also include the possibility for delivery of larger amounts of drugs or distinct classes of biomolecules (e.g., mRNA) by MV compared to Exo, as well as more successful targeting of MV cargo to the nuclear or cytosolic compartment, aspects that surely require further exploration.

It has become clear that MV have a multifaceted influence on cancer therapy, as summarized in Figure 3. They can not only serve as innovative tools for targeted drug delivery but should also be considered as important therapeutic targets when developing novel treatment strategies. Therefore, it will be highly advantageous to further elucidate the molecular mechanisms underlying MV formation and secretion in order to learn how MV biogenesis, especially in cancer cells, can be specifically inhibited or modulated therapeutically. However, at present, the lack of standardization, multicenter studies, and technical challenges, such as storage, production, quality control, and targeted delivery, still hamper the use of MV as diagnostic and therapeutic tools in cancer and must be overcome for them to be exploitable in the clinic.

## Figures and Tables

**Figure 1 ijms-21-05373-f001:**
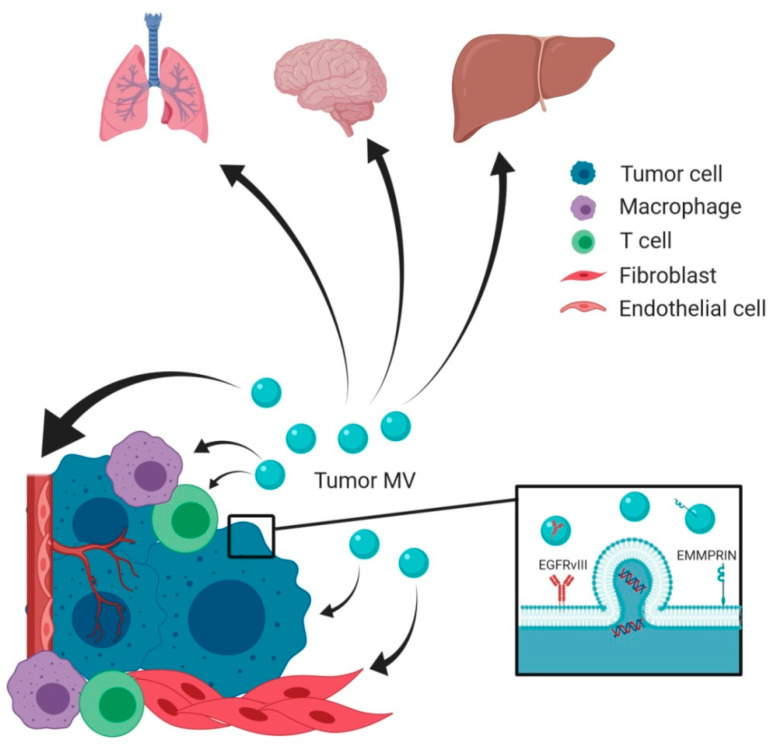
Microvesicles (MV) shaping the local tumor microenvironment (TME) and distant metastatic niches to promote tumor growth. On the one hand, MV shed from the plasma membrane of tumor cells act in an autologous way on other tumor cells by transferring oncogenic cargo and stimulating pro-tumorigenic signaling. On the other hand, MV are able to mediate heterologous cell–cell communication by acting on cells present within the TME, such as macrophages, T cells, fibroblasts, or endothelial cells. As a result, these cells become activated and change their phenotype to enhance tumor growth. Once the tumor has gained access to the circulation, tumor MV can be distributed throughout the body and mediate cargo transfer and cell–cell communication at distant sites. This induces the formation of pre-metastatic niches that offer a favorable environment for subsequent seeding of tumor cells in secondary organs.

**Figure 2 ijms-21-05373-f002:**
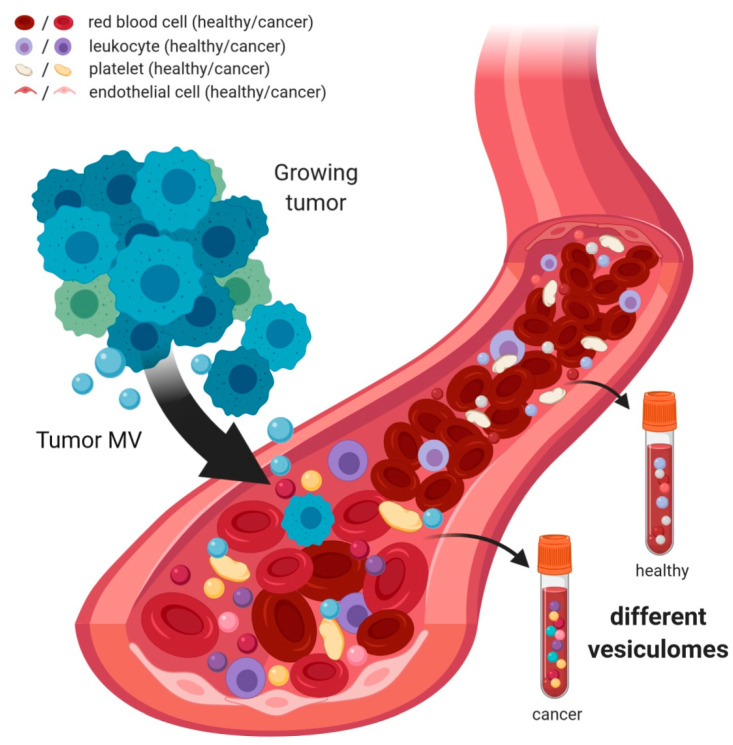
The microvesiculome in blood. The growth of a tumor is accompanied by secretion of circulating tumor cells (CTCs) as well as of tumor-derived soluble factors and MV. When encountering any of these components, benign blood cells can become activated and react by shedding their own, altered MV. Taken together, this results in changes in the composition as well as expression pattern of MV in the blood of cancer patients. Liquid biopsy-based MV sampling should therefore not only be focused on the detection of tumor MV, but also on alterations in non-tumor blood MV.

**Figure 3 ijms-21-05373-f003:**
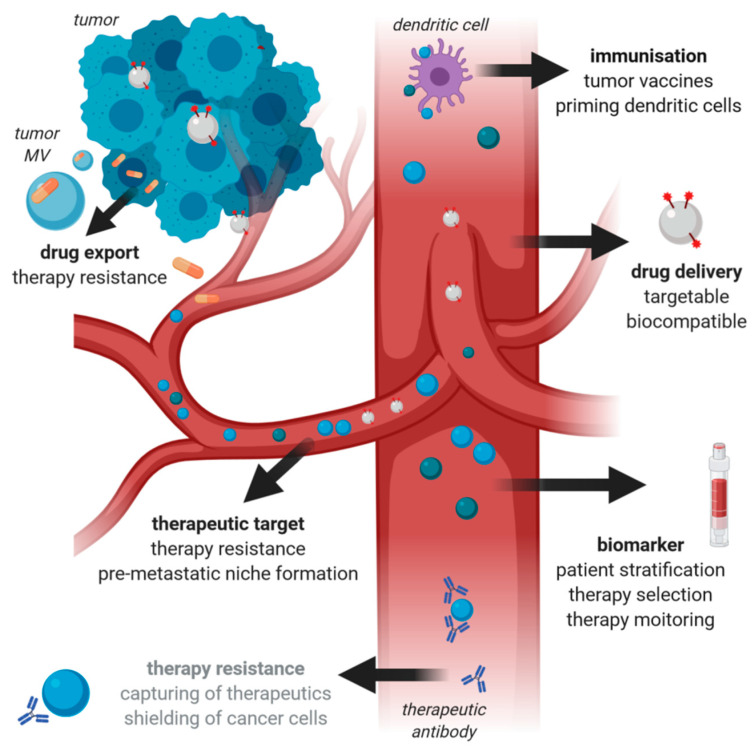
MV in cancer therapy. The figure summarizes the roles of MV in cancer therapy. Once a tumor has gained access to the circulation, circulating tumor MV in blood can be used as biomarkers in liquid biopsies. Moreover, they can mediate therapy resistance by capturing or exporting the administered anti-cancer drugs. Due to their role in therapy resistance and tumor microenvironment crosstalk resulting in pre-metastatic niche formation, they should be regarded as a novel therapeutic target. However, MV can also have a beneficial role in cancer therapy as they represent promising novel drug delivery systems with the benefits of high biocompatibility as well as opportunities for specific targeting and crossing of biological barriers. Vesicular functions, which have only been reported for endosomal-derived small exosomes (Exo) and lack experimental proof for MV, are highlighted in grey.

**Table 1 ijms-21-05373-t001:** Common tumor antigens detected on patient-derived MV.

Marker	Cancer Subtype	Method	Reference
CD13	Leukemia (AML/CML) Myelodysplastic syndrome	Flow cytometry	[91]
CD138	Multiple myeloma	Flow cytometry	[89]
CD19	Non-Hodgkin lymphoma Leukemia (CLL) Morbus Waldenström	Flow cytometry	[91]
CD30	Hodgkin lymphoma	Flow cytometry	[91]
CD38	Multiple myeloma	Flow cytometry	[91]
c-Met	Gastric cancer	Western blot	[84]
Caveolin-1	Prostate cancer	Flow cytometry	[120]
EGFR	Colorectal	Flow cytometry Western blot	[76] [117]
	GBM	Immunofluorescence	[94]
EGFRvIII	GBM	Immunofluorescence	[94]
EMMPRIN	Various	Flow cytometry Western blot	[11] [76]
	Colorectal cancer Lung cancer Pancreas carcinoma	Flow cytometry	[119]
	Gastric cancer	Western blot	[84]
EpCAM	Breast cancer Lung cancer Head and neck cancer Colorectal cancer Pancreatic cancer	Flow cytometry	[76] [119]
	GBM	Immunofluorescence	[94]
EpCAM/EMMPRIN	Colorectal cancer Lung cancer Pancreas carcinoma	Flow cytometry	[119]
EpCAM/ASGPR1	Hepatocellular carcinoma Cholangiocarcinoma	Flow cytometry	[118]
FAK	Breast cancer	Western blot	[87]
HepPar1	Hepatocellular carcinoma	Flow cytometry	[121]
Her2/Neu	Gastric cancer	Flow cytometry	[84]
	Colorectal cancer	Flow cytometry Western blot	[117]
MUC1	Breast cancer Pancreatic cancer	Flow cytometry	[86]
	Breast cancer	Flow cytometry Western blot	[11]
	Breast cancer Lung cancer Head and neck cancer	Flow cytometry	[76]

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
