# Peer review of "Microvesicles in Cancer: Small Size, Large Potential"

_ijms, 2020, doi:10.3390/ijms21155373_

Round 1

Reviewer 1 Report

Authors reviewed the MV in cancer focusing on protein. Authors emphasized in the importance of the methods of isolation of MV. The review would be useful to understand the MV in malignancies.

Auhors mentioned the differences of MV by the methods of purification. Please summarize the several methods to isolate MV, such as ultracentrifuges, sucrose cushion ultracenrifuges, and precipitations by antibody.

Authors refer the position paper of ISEV (ref 7). Please summarize the appropriate methods for the isolation of EV and the necessary markers for the validation of the appropriate methods,

To measure the protein levels in MV, the SRM/MRM is also promising. Discuss about it.

Urinary MV is also good sources analyzing the MV from urological cancer. Discuss about them.

Please add the tumor antigens in urological cancer in Table 1.

Author Response

Please see cover letter attached

Reviewer 2 Report

This review summarized function of microvesicles in the TME. It's nicely written with good drawings. It should be published.

Author Response

Please see cover letter attached

Round 2

Reviewer 1 Report

Authors revised the manuscript adequately.